# An adjustable algal chloroplast plug-and-play model for genome-scale metabolic models

**Gunvor Bjerkelund Røkke** [1], **Martin Frank Hohmann-Marriott** [1], **Eivind Almaas** [1,2] *

**1** Department of Biotechnology and Food Science, The Norwegian University of Science and Technology, Trondheim, Norway, **2** K. G. Jebsen Center for Genetic Epidemiology, The Norwegian University of Science and Technology, Trondheim, Norway

* eivind.almaas@ntnu.no

**Data Availability Statement:** All relevant data are within the manuscript and its Supporting Information files.

**Funding:** The Faculty of Natural Sciences (priority programme NTNU Ocean) at the Norwegian

## Abstract

The chloroplast is a central part of plant cells, as this is the organelle where the photosynthesis, fixation of inorganic carbon, and other key functions related to fatty acid synthesis and amino acid synthesis occur. Since this organelle should be an integral part of any genome-scale metabolic model for a microalgae or a higher plant, it is of great interest to generate a detailed and standardized chloroplast model. Additionally, we see the need for a novel type of sub-model template, or organelle model, which could be incorporated into a larger, less specific genome-scale metabolic model, while allowing for minor differences between chloroplast-containing organisms. The result of this work is the very first standardized chloroplast model, iGR774, consisting of 788 reactions, 764 metabolites, and 774 genes. The model is currently able to run in three different modes, mimicking the chloroplast metabolism of three photosynthetic microalgae–*Nannochloropsis gaditana*, *Chlamydomonas reinhardtii* and *Phaeodactylum tricornutum*. In addition to developing the chloroplast metabolic network reconstruction, we have developed multiple software tools for working with this novel type of sub-model in the COBRA Toolbox for MATLAB, including tools for connecting the chloroplast model to a genome-scale metabolic reconstruction in need of a chloroplast, for switching the model between running in different organism modes, and for expanding it by introducing more reactions either related to one of the current organisms included in the model, or to a new organism.

## Introduction

The chloroplast is a vital part of any plant cell. This is the organelle where the light reactions of photosynthesis occur [1–3], and also the organelle that hosts the Calvin-Benson cycle [2,4,5]. Since the Calvin-Benson cycle is the single most important pathway for fixation of inorganic carbon, the chloroplast is responsible for nearly 100% of the primary production carried out by plants, algae and cyanobacteria. This particular organelle also contains important pathways related to fatty acid biosynthesis [6,7], pigment synthesis [8,9] and *de novo* synthesis of several amino acids [10,11] which are of great importance to mammals, as we are not able to produce most of these ourselves [12,13].

University of Science and Technology (NTNU) founded the PhD position (GBR). Grant number is not available. This funder had no role in study design, data collection and analysis, decision to publish, or preparation of the manuscript.

**Competing interests:** The authors have declared that no competing interests exist.

**Abbreviations:** API, Application Programming Interface; DGDG, Digalactosyl diacylglycerol; DHA, Docosahexaenoic acid; EC, Enzyme Commission; FBA, Flux balance analysis; FNR, Ferredoxin NADP$^+$ reductase; KEGG, Kyoto Encyclopedia of Genes and Genomes; MATLAB, Matrix Laboratory; MGDG, Monogalactosyl diacylglycerol; NCBI, National Center for Biotechnology Information; PSI, Photosystem I; PSII, Photosystem II; PTOX, Plastid terminal oxidase; SQDG, Sulfoquinovosyl diasylglycerol.

As the chloroplast is an essential part of a plant cell, having a good mathematical representation of the chloroplast is also essential to any model describing the metabolism of a plant or algal cell.

Since DNA sequencing technology has seen great improvements in the later years, and new genomes are being sequenced at an increasing pace, the need to ease construction of metabolic models have also emerged. As a result of this, automated tools for genome annotation and model construction have been developed [14,15]. This has improved the time aspect of creating metabolic models for newly sequenced organisms, and also contributed to a standardization of the construction of metabolic models. However, some of the automatic model construction approaches that have been developed, do not provide the user with detailed insight into how the model is being constructed or how gaps in a draft metabolic network are filled. Since these tools are automated and purely based on the information available in databases, pathways and reactions that are not actually present in the organism might be introduced to the model, while other reactions might be missing if the enzymes responsible for catalysing them are organism-specific and not yet present in databases. As of yet, metabolic models assembled by an automated approach also give the user few opportunities to alter the resulting model.

As an alternative approach to ease model construction of chloroplast-containing cells, we have developed a standardized chloroplast model as a plug-and-play module that can be run in different organism modes. The chloroplast model is built on chloroplast-specific pathways found in other models of chloroplast metabolism, but the photosynthetic processes, which are often incorrectly represented in published metabolic models for photosynthetic organisms, have been carefully curated in this chloroplast model. This novel approach of developing an organelle-focused plug-and-play module that standardizes the relatively conserved chloroplast metabolism, also allows for the fine-tuning of minor metabolic differences between different organisms containing a chloroplast. Thus, we propose the approach of manually generating high-quality organelle template genome-scale metabolic modules as a new paradigm for improved automated reconstruction of eukaryotic genome-scale metabolic models. Additionally, we have developed several computational tools for connecting the chloroplast module to a pre-existing genome-scale metabolic model lacking a chloroplast, for changing organism-mode of the chloroplast, and for making it easy to introduce additional reactions to the model in general or to the organism-specific parts of it. The software allows new organism modes to be introduced.

## Materials and methods

### Reconstruction of the draft metabolic model

Previously published reconstructions of *Nannochloropsis gaditana* [16], *Chlamydomonas reinhardtii* [17] and *Phaeodactylum tricornutum* [18] were used as basis for our draft chloroplast model iGR774. The *N. gaditana* model was used as a base for most of the common chloroplast metabolic processes shared between the three algae, while the *C. reinhardtii* model and the *P. tricornutum* models were used for introducing organism-specific parts as a proof-of-principle.

To be able to distinguish between reactions common to all three organisms and strictly organism-specific reactions, the organism-specific reactions were given a tag indicating which organism the reaction in question is specific to. For *Nannochloropsis*, the organism specific tag is '@Nan', which was used as a prefix to the ID of the reaction in question (example: '@Nan_R05345_h'). For *Chlamydomonas* and *Phaeodactylum* specific reactions, '@Chl' and '@Pha' were used as organism specific reaction tags. These tags are used by the script developed to switch the model from one organism-mode to another in order to recognize organism-specific reactions, and determining the flux limits for these in the current organism-mode.

## Curation of the draft metabolic model

Since the aim of this work was to create a standardized model of chloroplast metabolism, the draft metabolic model was curated by standardizing pathways known from literature to be present in the chloroplasts of *Nannochloropsis*, *Chlamydomonas* and *Phaeodactylum*. Special emphasis was put on pathways involved in lipid synthesis, pigment synthesis and synthesis of animo acids, as these are groups of molecules that are currently attracting much interest from the academic community. Other pathways, for example the ones involved in synthesis of RNA and DNA, were omnitted, as these are not considered essential in the function of a metabolic chloroplast module. Certain reactions also needed to be introduced during reconstruction, and for this purpose, tools were developed to ease the import of new reactions to the model from KEGG (Kyoto Encyclopedia of Genes and Genomes) (see section on method development).

Since photosynthesis is an essential part of the chloroplast and its normal function, the photosynthetic electron transfers were also modelled in detail as part of the curation process. When modelling photosynthesis as part of a metabolic model of a photosynthetic organism, it has been common to describe each photosynthetic complex as one distinct reaction. In some reaction system reconstructions, additional processes related to photosynthetic activity, such as cyclic electron transport [19–20], the Mehler reaction [21] and plastoquinone re-oxidation by the enzyme Plastid Terminal Oxidase (PTOX) [22] has also been modelled as part of the photosynthetic activity.

To give a more detailed overview of the processes involved in photosynthesis, and to be able to assess electron fluxes, we modelled every electron transfer process occurring in the photosynthetic electron transport chain as one distinct reaction, yielding 33 photosynthetic reactions, instead of the usual 5 to 8.

Genetic information was also added to the chloroplast model (see the *genes* section for more information). Finally, as part of the curation process, the maximum and minimum fluxes of certain reactions were restricted (for a complete list of restricted reactions, see S1 Table). This was necessary to make the model produce a realistic set of metabolic fluxes, as the chloroplast model is a module meant to be fused with an exo-model. Since the chloroplast module is developed to be a part of another model in need of a chloroplast, the chloroplast module includes a set of cytosolic metabolites, in addition to several exchange reactions responsible for transporting these metabolites between the chloroplast and the cytoplasm. When the chloroplast module is coupled to an exo-model, the script being responsible for the fusion of the models will localize the exo-model version of the exchange metabolites of the chloroplast, and the names of the chloroplast exchange metabolites will be changed into the namespace of the exo-model, to ensure a seamless fusion of the two models. If one or more of the chloroplast exchange metabolites are not present in the exo-model, the user will be notified to aid in the debugging in case the combined model does not run. When the chloroplast model operates on its own, no cytosolic reactions are present to generate or consume the exchanged metabolites. When run as a standalone model, the cytosolic metabolites are therefore allowed to be imported or exported freely from the model's external environment. These reactions (tagged 'B_' for 'border reaction' in the chloroplast model) are automatically deleted when the chloroplast model us used as a module that is fused with an exo-model.

Many exchange reactions are involved in exchange of carbon compounds. If these exchange reactions were allowed to be run in both directions without constraints, the chloroplast model, when tested on its own, would be perfectly able to produce / import all the compounds it is supposed to produce, without running photosynthesis and photosynthetic carbon fixation.

## Genes

The chloroplast model was supplied with genetic information, mainly from KEGG and NCBI (National Center for Biotechnology Information). KEGG reactions are sometimes accompanied by information about the protein catalysing them in different organisms. When a KEGG reaction was imported to one of the organism-specific parts of the model by the script pick-KEGGrx (described in S1 Text), the KEGG API (Application Programming Interface) and MATLAB (Matrix Laboratory, MathWorks, USA) was used to search for the EC (Enzyme Commission) number of the enzyme(s) catalysing the KEGG reaction in question, and for each KEGG number found, KEGG API and MATLAB was again used to search for *Nannochloropsis*, *Chlamydomonas* or *Phaeodactylum* specific genes catalysing the enzyme in question. If genes were found, they were added to the chloroplast model for the reaction in question.

For adding genetic information to the photosynthetic electron transfers carried out by one of the three multisubunit protein complexes of the photosynthetic electron transport chain; Photosystem II (PSII), the cytochrome $b_6f$ complex ($b_6f$) or Photosystem I (PSI), or the reaction carried out by the proton-driven complex ATP synthase, the NCBI Protein database was used. A list of all gene–protein associations in *Nannochloropsis* was downloaded, and the list was scanned for the protein subunits of PSII, the cytochrome $b_6f$ complex, PSI or ATP synthase. Genes for several subunits of all four protein complexes were identified. These were added to the model with an 'AND' relationship for all electron transfers occurring in the complex in question.

## Chloroplast biomass reaction

When the chloroplast model is merged with an exo-model, the exo-model's biomass reaction will be the prevailing biomass reaction, and might be modified manually by the user to account for the addition of a chloroplast.

However, to be able to test the chloroplast model's ability to produce the many important metabolites it should be able to produce, a separate biomass reaction had to be added to the chloroplast model. This 'chloroplast biomass' reaction is deleted by the script plugging the chloroplast model into an exo-model, but when the chloroplast model is tested on its own, this is the reaction that will be optimized.

The most important metabolite classes produced by the chloroplast are the membrane glycolipids monogalactosyldiacylglycerol (MGDG), digalactosyldiacylglycerol (DGDG) and sulfoquinovosyldiacylglycerol (SQDG) [11], the amino acids lysine, methionine, threonine, phenylalanine, tyrosine, tryptophan, leucine, isoleucine and valine [11,13], an organism-specific pool of pigments [8,23] in addition to ATP and NADPH, which are the end-products of the photosynthetic electron transport chain.

The chloroplast specific biomass reaction is therefore composed of the chemical equations $ATP + H_2O \rightarrow ADP + P_i$ ($P_i$ denoting inorganic phosphate), $NADPH \rightarrow 2\,e^- + NADP^+ + H^+$, and $O_2 + 4\,H^+ + 4\,e^- \rightarrow 2\,H_2O$, the latter equation being present in the biomass solution in order to balance out the electrons between the left and the right side of the equation. The organism-specific chloroplast-produced fractions of membrane lipids, proteins and pigments are also consumed in the chloroplast biomass reaction. Keeping electron balance in mind, and accounting for the fact that the chloroplast usually produces more ATP than NADPH, the following reaction was used as chloroplast biomass:

$$10\ ATP + 5H_2O + 5\ NADPH + 5\ H^+ + 2.5\ O_2 + 1\ CPr + 1\ CML + 1\ CPi$$
$$\rightarrow 10\ ADP + 10\ P_i + 5\ NADP^+ \tag{1}$$

In the chloroplast biomass equation above, CPr denotes chloroplast proteins, CML denotes chloroplast membrane lipids and CPi denotes chloroplast pigments.

The composition of the chloroplast protein fraction, the membrane lipid fraction and the pigment fraction consumed in the chloroplast biomass reaction are organism-specific, and in practice, one organism-specific biomass reaction is present for every organism-mode included in the chloroplast model. Currently, one biomass reaction is present for each of the three organisms *Nannochloropsis*, *Chlamydomonas* and *Phaeodactylum*, but the chloroplast model can be expanded to be able to describe the metabolism of more organisms, and for every new organism mode that is added, one biomass reaction must also be added to describe the chloroplast production of the organism in question. When switching between organism modes, the biomass reaction is also changed accordingly, unless the chloroplast model is coupled to an exo-model. Then the biomass reaction of the exo-model will be the reaction to be optimized.

The organism-specific biomass components accounting for membrane lipids, protein and pigment production were modelled according to stoichiometric values found in literature. For *Nannochloropsis*, the ratios between the different fatty acids in different membrane lipids were taken from Vieler et al. 2012 [11], while the ratio between MGDG, DGDG and SQDG in the cell membrane was taken from Li et al. 2014 [24]. The ratios between different fatty acids in the different classes of membrane lipids in *Chlamydomonas* were taken from Suh et al. 2015 [25], while the ratio between the different lipid classes was taken from Boudière et al. 2014 [26].

In *Phaeodactylum*, the ratios between fatty acids in the different membrane lipids were taken from Tonon et al. 2002 [27], and the ratio between different membrane lipids was found in Arao et al. 1987 [28].

The mean composition of the different amino acids in an average protein was taken from Xiao et al. 2013 [29] for *Nannochloropsis*, from Boyle & Morgan, 2009 [30] for *Chlamydomonas* and from Brown, 1991 [31] for *Phaeodactylum*.

The pigment composition in *Nannochloropsis* was found in Lubián et al. 2000 [8], and in Eichenberger et al. 1986 [23] for *Chlamydomonas*. *Phaeodactylum* contains some pigments that are not found in the two other organisms, for example fucoxanthin, diatoxanthin, diadinoxanthin and chlorophyll *c* in addition to chlorophyll *a* [32]. The production of these require specific pathways that could be added to the model in the future. The detailed composition of the chloroplast biomass components 'chloroplast proteins', 'chloroplast membrane lipids' and 'chloroplast pigments' is described in S2 Text.

## Constraint-based linear optimizations

Mathematically, the metabolic network of iGR774 is represented by an *M x N* sized stoichiometric matrix S, where *M* denotes the number of metabolites, and *N* denotes the number of reactions. A positive stoichiometric coefficient $s_{ij}$, thus indicate that s molecules of the $i^{th}$ metabolite is produced in the $j^{th}$ reaction. A negative stoichiometric coefficient indicates consumption of the metabolite instead of production. Flux balance analysis (FBA) was used for optimization of the model under criteria of steady-state, represented by the linear problem

$$S \cdot v = 0,$$

where ν is the flux vector that best solves the optimization problem.

For a list of reactions that were restricted, and the reason for restricting the reaction in question, see S1 Table.

FBA optimizations were performed using MATLAB and COBRA Toolbox [33] in combination with the Gurobi optimizer (version 8.1.0, Gurobi Optimization Inc., Houston, Texas).

## Method development

The development of a sub-model meant to be plugged into an exo-model, and its ability to run in different organism-modes, is to our knowledge a new concept. For this reason, the model reconstruction work was accompanied by the development of a new toolbox of scripts. The newly developed scripts were written in MATLAB code, and will ease the process of working with a compartment model with the ability to be connected to another model.

When constructing the model, certain pathways specific to one of the organisms included in the chloroplast model had to be transferred from other models. An example of such a pathway was the eyespot which was transferred from the *Chlamydomonas* model published by Imam et al. in 2015 [17]. A specific script was written for the purpose of transferring reactions. The script simultaneously translate the metabolites into the namespace of the exo-model, and also checks that the reaction names are not already present in the chloroplast model.

Certain reactions were introduced to the chloroplast model from KEGG. Another script was therefore written to enable the import of KEGG reactions to the model. This script also translates the metabolites of the reaction in question to the chloroplast model namespace.

The chloroplast model currently contains 788 reactions, but it could still be further expanded, both by adding reactions to the organism modes already included in the model and by including new organism modes. The scripts for adding reactions, both from other models, from KEGG and with inputs from the user could ease the future process of expansion of the chloroplast model.

In order to ease the process of making the model more detailed, scripts were also developed for adding genetic information to the model, both to existing reactions, and to new ones.

In order to identify gaps in pathways, or to verify that a certain pathway (or the entire model, for that matter) is connected, several tools were developed. One script allows the user to follow a path of metabolites from a certain metabolite, treating the metabolite in question as either a substrate or a product. In order to check if a pathway or an entire model is connected, scripts were also written to transfer a model or part of a model to a format that can be imported into cytoscape, enabling the user to inspect the model visually (metabolic models can also be visualized and analysed by the newly published tool ModelExplorer [34]).

Scripts were also written to plug the chloroplast model into an exo-model, and to change organism-mode of the chloroplast model.

When plugging the chloroplast model into an exo-model, the metabolite names and IDs of the chloroplast metabolites are first translated to the namespace used by the exo-model, and the basis for comparison between the metabolites of the two models are KEGG IDs. Several models do, however, not contain KEGG IDs for their metabolites, and this problem we solved by writing two scripts for supplying a model with metabolite KEGG IDs. Since a metabolic model can include several thousand metabolites, one of these scripts is running in a very automatic manner, requiring little input from the user, while the other requires more input from the user, and is meant to complement the first script.

Last but not least, several general scripts were written aiming at making it easier to work with a metabolic model in matlab, and also to display information about enzymes, metabolites and reactions from KEGG in MATLAB.

To access the newly developed tools, see S2 File, and for a detailed description of the tools, see S1 Text.

## Results and discussion

### Summary of reconstructed chloroplast model

The result of the reconstruction of chloroplast metabolism was the iGR744 model (S1 File) containing 788 reactions, 764 metabolites and 774 genes. The reactions of the model belong to one or more of 71 subsystems (the number of reactions affiliated with the most important overall subsystems groups is shown in Fig 1).

There were 110 genes that were shown to be critical for biomass production. Knockouts of 107 of these genes completely restricted the chloroplast model from producing biomass, while knockouts of 3 genes decreased the rate of biomass production. A list of the critical genes can be found in S2 Table.

There were 702 of the reactions included in the model that proved to be insensitive to organism mode, while 86 reactions were shown to be organism-specific.

There are 16 *Nannochloropsis* specific reactions in the model. These are mainly connected to synthesis of pigments that are only present in *Nannochloropsis*, and not in the other two organisms contained in the model. There are 50 reactions that are *Chlamydomonas* specific. Most of these are reactions occurring in the *Chlamydomonas* eyespot [35], which is an organelle *Nannochloropsis* and *Phaeodactylum* do not possess. There are also 16 reactions that are *Phaeodactylum* specific, and four organism-specific reactions are shared between *Phaeodactylum* and *Chlamydomonas*, as both of these organisms have the possibility of producing the fatty acid docosahexaenoic acid (DHA), although in relatively small quantities [25]. *Chlamydomonas* in addition produces C18:3, while this lipid is not produced to a large extent in *Nannochloropsis* and *Phaeodactylum* [11, 25, 27]. Each specific organism mode also includes reactions for generating MGDG, DGDG, SQDG, proteins and pigments. In addition, MGDG, DGDG and SQDG are combined into an organism-specific chloroplast membrane lipid component, where the different membrane lipids are added according to the correct organism-specific ratio. The exact composition of the organism-specific biomass components is given in S2 Text.

The chloroplast model is able to produce biomass running in both *Nannochloropsis* mode, *Chlamydomonas* mode, and *Phaeodactylum* mode. The growth rates are shown in Table 1.

As Table 1 show, the chloroplast growth rates are not the same when running in different organism modes. This is due to the fact that all the three possible organism modes currently introduced to the model have their own biomass reaction, adjusted to the chloroplast metabolism of the organism in question. The growth rates are similar when running the chloroplast in *Nannochloropsis* mode and *Chlamydomonas* mode, while the growth rate is higher when the chloroplast is run in *Phaeodactylum* mode. The increased growth rate for the *Phaeodactylum* mode might be due to the fact that *Phaeodactylum* has a specific pool of pigments, while the pathways responsible for producing these pigments have not yet been added to the model. The pigment pool included in the *Phaeodactylum* biomass reaction thus only contains chlorophyll *a*, which puts less requirements on the production pathways of the chloroplast compared to when it is running in *Nannochloropsis* or *Chlamydomonas* mode.

### Photosynthesis

In metabolic models describing photosynthetic organisms, photosynthesis itself is usually modelled using very few reactions. The photosynthetic machinery consists of the three electron transferring complexes photosystem II, the cytochrome $b_6f$ complex and photosystem I. These transfer electrons, which eventually are used to regenerate NADPH from $NADP^+$ by the enzyme Ferredoxin $NADP^+$ reductase (FNR) after increasing the energy of the electrons using sunlight. Photosynthetic electron transport does not only lead to the generation of NADPH. It

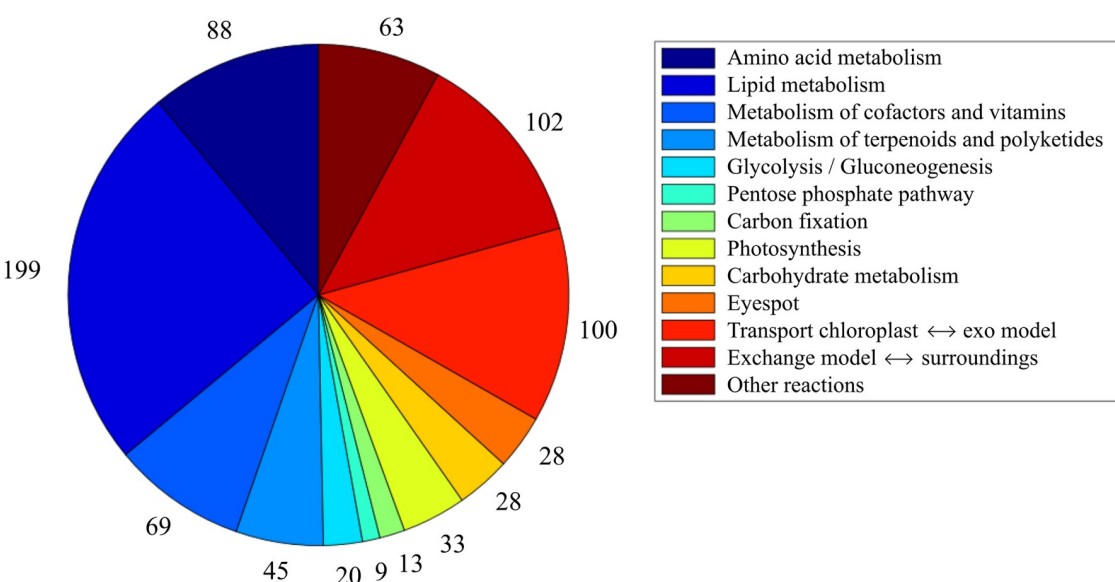

**Fig 1. The number of chloroplast reactions affiliated with each major subsystem group.**

also pumps protons across the thylakoid membrane of the chloroplast. When photosynthesis is running normally, the concentration of protons in the thylakoid lumen is higher than in the stroma, due to PSII and the cytochrome $b_6f$ complex. This imbalance in proton concentration on the two sides of the thylakoid membrane is the driving force for ATP synthesis by the fourth protein complex, ATP synthase.

When creating metabolic models describing photosynthetic algae, it has been common to describe each electron-transporting complex as one reaction. In addition, some models contain reactions describing processes involved in regulating photosynthesis, such as cyclic electron transport [36], plastoquinone oxidation by Plastid Terminal Oxidase (PTOX) [37] and the Mehler reaction [38].

Photosynthesis is usually described by at most 10 reactions (The *Phaeodactylum tricornutum* model by Levering et al. published in 2016 [18] is an example). In several models (for example the *Chlamydomonas reinhardtii* model by Imam et al. published in 2015 [17], the *Nannochloropsis gaditana* model by Shah et al. published in 2017 [16], and the *Nannochloropsis salina* model by Loira et al. published in 2017 [39]), the reactions describing the electron transferring complexes in photosynthesis are in addition not correct with regard to proton pumping across the thylakoid membrane, which will affect the ratio of ATP-production to NADPH production.

The one-reaction-per-complex way of modelling photosynthesis makes it a black box. We therefore modelled every electron transfer occurring in the three major photosynthetic protein complexes, PSII, PSI and the $b_6f$ complex, making the electron-transporting fluxes more

**Table 1. Growth rates of the chloroplast model simulated in *Nannochloropsis* mode, *Chlamydomonas* mode and *Phaeodactylum* mode, respectively.**

| Organism mode | Growth rate [mmol gDW$^{-1}$ h$^{-1}$] |
|---|---|
| *Nannochloropsis* | 0.0318 |
| *Chlamydomonas* | 0.0315 |
| *Phaeodactylum* | 0.0455 |

transparent. 10 of our modelled electron transfer reactions occur within PSII, 7 occur within the cytochrome $b_6f$ complex, 8 within PSI, while 3 are related to re-generation of NADPH by FNR. One reaction describes the re-generation of ATP from ADP by ATP synthase, two reactions are related to plastoquinone transport, while one last photosynthetic reaction describes cyclic electron transport around PSI, which uncouples the ratio between ATP and NADPH. The reconstruction of photosynthesis is shown in Fig 2.

The NADPH and ATP molecules generated as a result of the photosynthetic electron transport are largely used by the cell to fix carbon in the Calvin-Benson cycle [2,4,5]. An interesting property of a chloroplast model with a special focus on photosynthesis would therefore be how the relationship between photon usage by photosynthesis and carbon uptake rate in the Calvin-Benson cycle affects cell growth.

A phenotype phase plane showing this relationship is shown in Fig 3.

When optimizing the production of chloroplast biomass, the optimal flux through the RuBisCO-driven reaction responsible for $CO_2$ fixation, the general import reaction for photons used by photosynthesis, and for the reaction partitioning photons to PSII are 125 mmol $gDW^{-1} h^{-1}$ (DW represents dry weight), 1000 mmol $gDW^{-1} h^{-1}$ and 500 mmol $gDW^{-1} h^{-1}$, respectively. When running the chloroplast model, all photosynthetic reactions runs at their maximum allowed flux in all organism modes, restricted by the upper limit of photon import, which is set to 1000 mmol $gDW^{-1} h^{-1}$.

The RuBisCO-driven reaction responsible for fixing $CO_2$ is part of the Calvin-Benson cycle. This particular pathway is costly for the cells to keep running energy-wise, which is why it is functionally coupled to the photosynthetic electron transport chain.

When the Calvin-Benson cycle is running normally, it needs 3 molecules of $CO_2$, 9 molecules of ATP and 6 molecules of NADPH to produce one single molecule of glyceraldehyde-3-phosphate or dihydroxyacetonephosphate. This ratio of demand for ATP and NADPH is fixed in the Calvin-Benson cycle, but the chloroplast's need for the two energy carriers in general is not, as the general metabolism can be influenced by factors beyond the cell. The chloroplast therefore needs to be able to tune the production ratio of ATP to NADPH. This is done

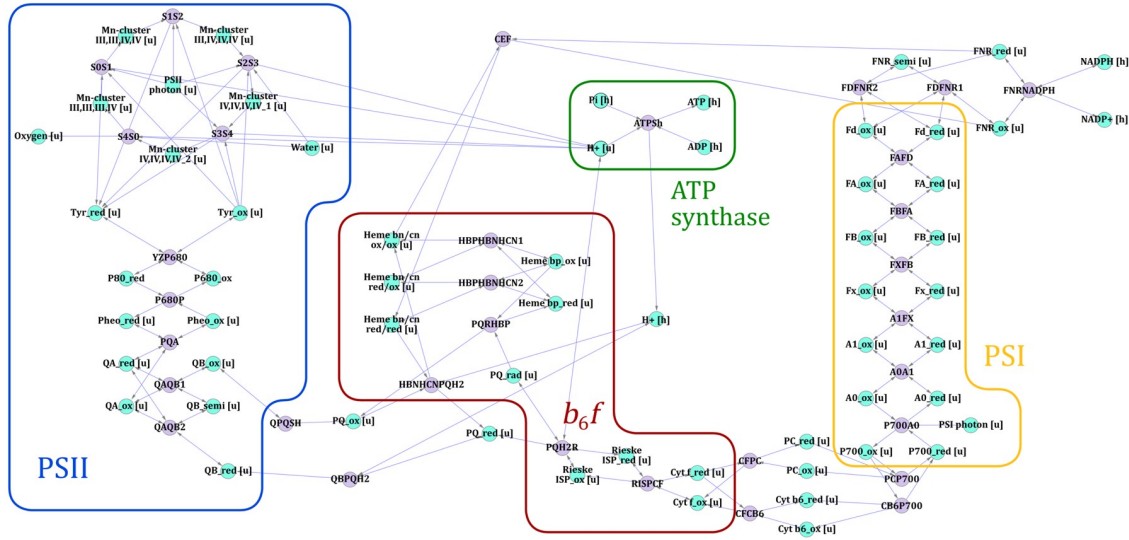

**Fig 2. Representation of photosynthesis in the chloroplast model.** The areas marked PSII, $b_6f$, PSI and ATP synthase shows where in the photosynthetic electron transport chain the different electron transfers and reactions are taking place. Light blue nodes represent metabolites, while purple nodes represent reactions.

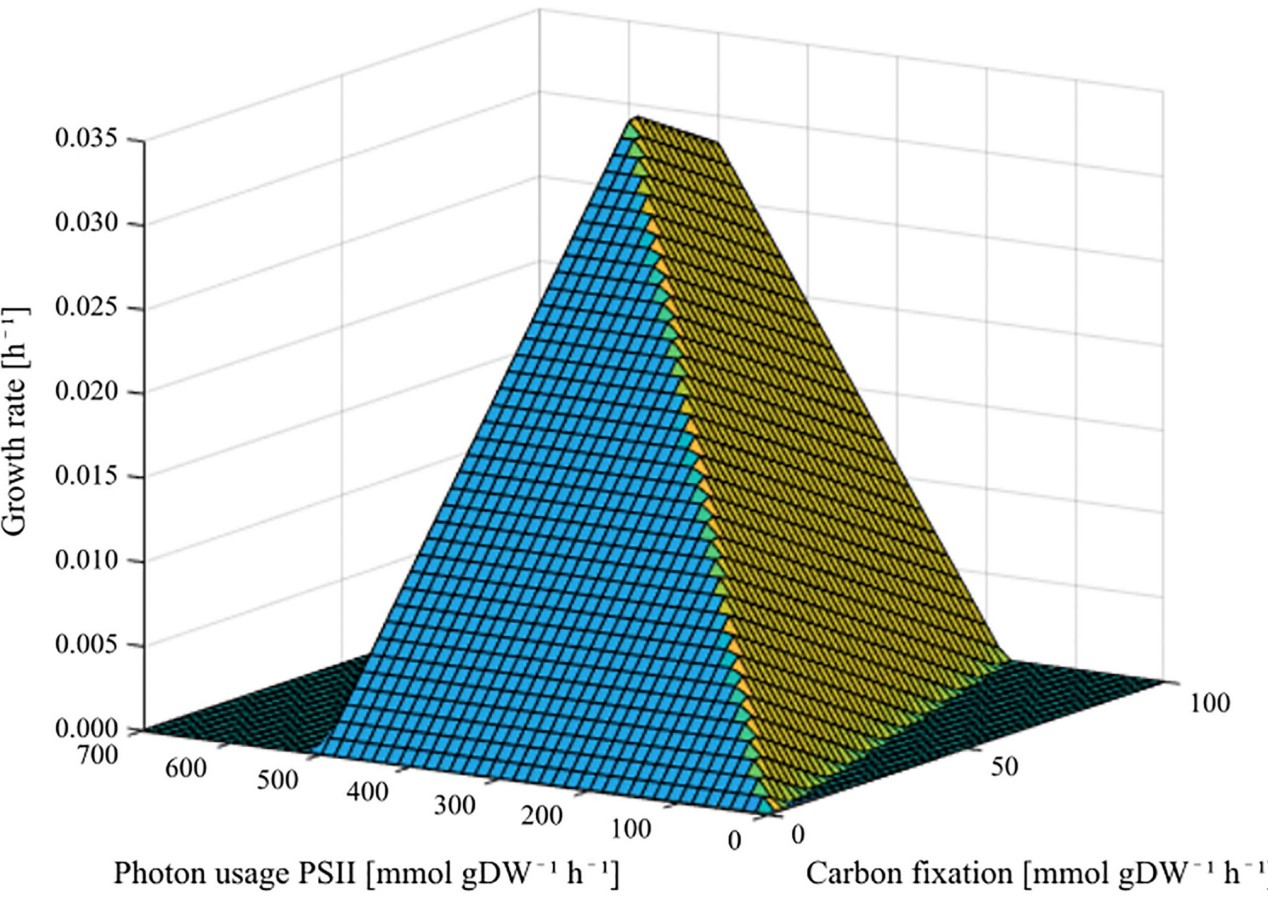

**Fig 3. Phenotype phase plane showing how the fluxes of carbon fixation (NanoG0589) and photon usage by PSII (PSII_photon) affect the flux of the chloroplast biomass function.**

by performing cyclic electron transport around PSI [36]. Cyclic electron transport, as the name implies, sends electrons on a cyclic journey around PSI. When an electron is 'recycled' in this manner, they are sent back to the plastoquinone pool, which is a pool of mobile electron carriers shuttling between PSII and the cytochrome $b_6f$ complex. Both the plastoquinone transport and the mechanism of the $b_6f$ complex contribute to the pumping of protons from the stroma into the thylakoid lumen. Cyclic electron transport thus adds to the proton gradient over the thylakoid membrane, while temporarily keeping electrons from being used to regenerate NADPH. Cyclic electron transport is thus the photosynthetic electron transport chain's way of tuning the ATP:NADPH production ratio.

In the biological world, rate of cyclic electron transport is mediated by certain protein complexes [40], which are regulated by transcription factors; and by the partitioning of light-energy between PSII and PSI [41]. In optimizations of metabolism by FBA, however, the only regulating force is mathematics, and finding the most beneficial set of fluxes solving the linear problem in question. A mathematical model of metabolism might therefore occasionally take shortcuts. This proved to be the case in photosynthesis. Mathematically speaking, it seems to be beneficial for the model to optimize the photosynthetic ATP production, while using non-photosynthetic reactions to create NADPH. When the upper flux limit for the reaction carrying out cyclic electron transport is allowed to run unlimited, the optimal flux is 500 mmol gDW$^{-1}$ h$^{-1}$, which corresponds to a recycling of all photosynthetically generated electrons.

The motivation for building metabolic models, is to describe the metabolism of an actual cell. Even though it seems to be beneficial mathematically speaking to run cyclic electron transport at full speed, it does not describe what goes on in an actual cell, and the rate of cyclic electron transport have therefore been restricted with regard to the ATP and NADPH demands of the Calvin-Benson cycle.

To find the upper bound for cyclic electron transport best matching the demands of the Calvin-Benson cycle, a mini-model was built, consisting only of the photosynthetic electron transport chain and the Calvin-Benson cycle. This mini-model was optimized for production of dihydroxyacetonephosphate, and since it did not contain transport reactions for ATP, ADP, inorganic phosphate, NADPH or NADP$^+$, the photosynthetic electron transport chain was forced to produce ATP and NADPH in the exact ratio needed for fuelling the Calvin-Benson cycle. The set of fluxes resulting from this optimization can be found in S3 Table.

The mini-model showed that the photosynthetic electron transport chain produces the exact ATP:NADPH ratio needed by the Calvin-Benson cycle, without having to run cyclic electron transport. This process have therefore been completely restricted in the chloroplast model, but the upper bound of cyclic electron transport can of course be changed for the purpose of analysing the impact of cyclic electron transport on varous aspects of the chloroplast metabolism.

### Lipid synthesis

Since especially *Nannochloropsis* is a potential target for industrial lipid production, the ability of the *Nannochloropsis* chloroplast to produce lipids was explored. Even though specific classes of lipids might be of interest when using a microalga as a commercial lipid-producer, the chloroplast model contains a combined pool of important chloroplast-produced lipids, and when exploring the chloroplast model's ability to produce *Nannochloropsis*-specific lipids, the production of this combined lipid pool was optimized.

Metabolism is a continuous process, and all the reactions are running simultaneously. Even so, certain reactions depend on the presence of other reactions to run. This is the case with lipid synthesis, which depends on carbon fixation by the Calvin-Benson cycle, which again depends on photosynthesis to fill its requirement for ATP and NADPH. The ability to produce lipids was therefore explored, using carbon fixation by RuBisCO and photon usage by PSII as control reactions. The result can be seen in Fig 4.

Lipids could be produced as a maximum growth rate of 0.0819 mmol gDW$^{-1}$ h$^{-1}$. The corresponding fluxes for photon usage by PSII and CO$_2$ fixation by RuBisCO were found to be 500 mmol gDW$^{-1}$ h$^{-1}$ and 150.038 mmol gDW$^{-1}$ h$^{-1}$, respectively. Furthermore, the phenotype phaseplane presented in Fig 4 nicely shows the lipid production's dependency on both photosynthesis and the Calvin-Benson cycle, and the cycle's dependency on photosynthesis, while photosynthesis is not dependent on either the Calvin-Benson cycle nor lipid production.

### Conclusions

We have constructed the chloroplast metabolic model iGR774, containing 788 reactions, 764 metabolites, and 774 genes. This is the first metabolic model developed as a plug-and-play module to represent a high-quality reconstruction of the metabolism of a single organelle, and the module may be run in different organism modes. The iGR774 model is thus intended to be incorporated into another genome-scale metabolic model in need of a chloroplast organelle. We suggest the plug-and-play concept as a new paradigm for generating high-quality genome-scale metabolic models for eukaryotic organisms. Here, a general challenge in generating automated model reconstructions is associated with reaction localization to the different

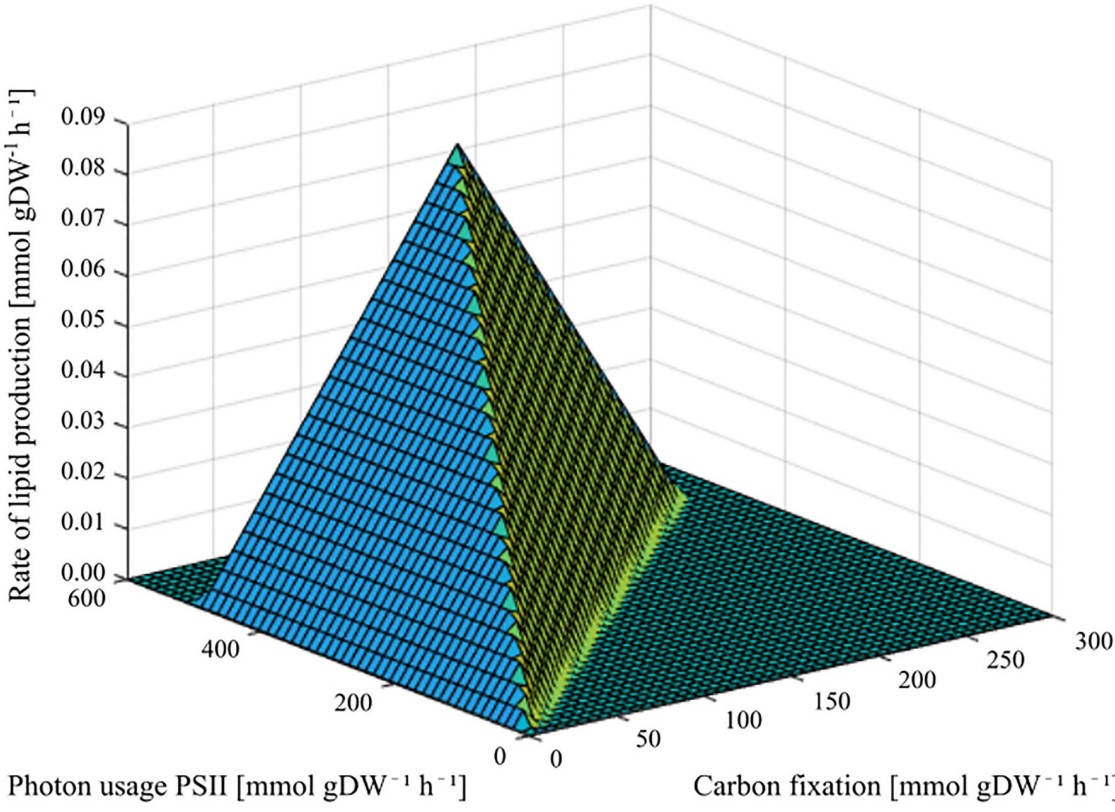

**Fig 4. Phenotype phaseplane showing the dependency of lipid synthesis on photosynthesis (photon usage by PSII used as control reaction) and the Calvin-Benson cycle (CO$_2$ fixation by RuBisCO used as control reaction).**

subcellular compartments. By instead developing a set of high-quality organelle modules, we propose that automated reconstruction frameworks may significantly improve their fidelity by using the modules as template reaction sets. A conceptually related approach to the use of template reaction sets is seen in the differentiation of human genome-scale metabolic reconstructions [42, 43] into specific tissue types, or the approaches to develop microbial genome-scale metabolic models for individual species and communities [44].

During the construction of this model, we have especially focused on photosynthesis, since this is arguably the most important biological process occurring in nature, and also a process that is largely neglected, and sometimes even represented incorrectly in genome-scale metabolic reconstructions of photosynthetic organisms. Photosynthetic electron transport fuels the Calvin-Benson cycle, which is responsible for the ability of microalgae and higher plants to fix carbon in the form of CO$_2$. This influx of carbon makes photosynthetic organisms able to produce large amounts of lipids, and thus makes them relevant targets for industrial lipid production. The chloroplast model can therefore be used to explore and compare modes of lipid production in the currently included microalgae modes, *Nannochloropsis*, *Phaeodactylum* and *Chlamydomonas*.

We have also developed software tools for incorporating the chloroplast model as a module into another metabolic model in need of a chloroplast, for changing the organism mode of the chloroplast model, and for further expanding the model, either by adding organism-specific reactions for one of the organisms already included, or by adding a new organism to the model.

## Supporting information

**S1 File. Model structure.** XML file containing the iGR774 model.
(XML)

**S2 File. Modelling tools developed alongside with the iGR774 model.** Zip-file containing modelling tools developed for MATLAB. The individual tools are described in S1 Text.
(ZIP)

**S1 Text. Descriptions of modelling tools.** Detailed description of the modelling tools included in S2 File.
(DOCX)

**S2 Text. Composition of chloroplast biomass components.** Detailed composition of the protein, membrane lipids and pigment fractions included in the chloroplast biomass component.
(DOCX)

**S1 Table. Restricted reaction.** List of reactions with restricted upper or lower bounds, including reason for the restriction.
(XLSX)

**S2 Table. Critical genes.** List of genes that proved to be critical to normal growth of the iGR774 model.
(XLSX)

**S3 Table. Photosynthesis and Calvin-Benson cycle fluxes.** List of fluxes resulting from optimization of a mini-model consisting of only the photosynthetic electron transport chain and the Calvin-Benson cycle.
(XLSX)

## Author Contributions

**Conceptualization:** Gunvor Bjerkelund Røkke, Martin Frank Hohmann-Marriott, Eivind Almaas.

**Data curation:** Gunvor Bjerkelund Røkke.

**Formal analysis:** Gunvor Bjerkelund Røkke.

**Funding acquisition:** Martin Frank Hohmann-Marriott.

**Investigation:** Gunvor Bjerkelund Røkke.

**Methodology:** Gunvor Bjerkelund Røkke, Eivind Almaas.

**Project administration:** Martin Frank Hohmann-Marriott, Eivind Almaas.

**Resources:** Martin Frank Hohmann-Marriott, Eivind Almaas.

**Software:** Gunvor Bjerkelund Røkke, Eivind Almaas.

**Supervision:** Martin Frank Hohmann-Marriott, Eivind Almaas.

**Validation:** Gunvor Bjerkelund Røkke, Eivind Almaas.

**Visualization:** Gunvor Bjerkelund Røkke.

**Writing – original draft:** Gunvor Bjerkelund Røkke.

 

**Writing – review & editing:** Gunvor Bjerkelund Røkke, Martin Frank Hohmann-Marriott, Eivind Almaas.

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
