## [Decision Letter · Decision Letter 0]

9 Dec 2019

PONE-D-19-26969

An adjustable algal chloroplast plug-and-play model for genome-scale metabolic models

PLOS ONE

Dear Dr. Bjerkelund Røkke,

Thank you for submitting your manuscript to PLOS ONE. After careful consideration, we feel that it has merit but does not fully meet PLOS ONE’s publication criteria as it currently stands. Therefore, we invite you to submit a revised version of the manuscript that addresses the points raised during the review process.

We would appreciate receiving your revised manuscript by Jan 23 2020 11:59PM. To enhance the reproducibility of your results, we recommend that if applicable you deposit your laboratory protocols in protocols.io, where a protocol can be assigned its own identifier (DOI) such that it can be cited independently in the future. For instructions see: http://journals.plos.org/plosone/s/submission-guidelines#loc-laboratory-protocols

We look forward to receiving your revised manuscript.

Kind regards,

Andrew Webber

Academic Editor

PLOS ONE

Journal Requirements:

1) When submitting your revision, we need you to address these additional requirements.Please ensure that your manuscript meets PLOS ONE's style requirements, including those for file naming. The PLOS ONE style templates can be found at

2) PLOS requires an ORCID iD for the corresponding author in Editorial Manager on papers submitted after December 6th, 2016. Please ensure that you have an ORCID iD and that it is validated in Editorial Manager. To do this, go to ‘Update my Information’ (in the upper left-hand corner of the main menu), and click on the Fetch/Validate link next to the ORCID field. This will take you to the ORCID site and allow you to create a new iD or authenticate a pre-existing iD in Editorial Manager. Please see the following video for instructions on linking an ORCID iD to your Editorial Manager account: https://www.youtube.com/watch?v=_xcclfuvtxQ

3)  Thank you for stating the following financial disclosure:

No. The funders had no role in study design, data collection and analysis, decision to publish, or preparation of the manuscript.

a) Please provide an amended Funding Statement that declares *all* the funding or sources of support received during this specific study (whether external or internal to your organization) as detailed online in our guide for authors at http://journals.plos.org/plosone/s/submit-now.  

b) Please state what role the funders took in the study.  If any authors received a salary from any of your funders, please state which authors and which funder. If the funders had no role, please state: "The funders had no role in study design, data collection and analysis, decision to publish, or preparation of the manuscript."

Additional Editor Comments:

The manuscript needs some very significant revisions as suggested by the comments below. Hopefully, these comments/suggestions can be used to constructively update the manuscript. It would be very valuable to include import of nuclear encoded components into the model.

Reviewers' comments:

Reviewer's Responses to Questions

**Comments to the Author**

1. Is the manuscript technically sound, and do the data support the conclusions?

Reviewer #1: Partly

2. Has the statistical analysis been performed appropriately and rigorously? 

Reviewer #1: I Don't Know

3. Have the authors made all data underlying the findings in their manuscript fully available?

Reviewer #1: Yes

4. Is the manuscript presented in an intelligible fashion and written in standard English?

Reviewer #1: No

5. Review Comments to the Author

Reviewer #1: The manuscript as written provides little new insight into the metabolic fluxes/metabolism taking place within chloroplasts of algae. A lot of statements are obvious or text book level. For example line 359-372, 410-431, 489-493 are all basic knowledge of photosynthesis.

The authors should add import of substrates from the cytosol to the model. This is how metabolism actually takes place within algae. This also would allow simulation of dark conditions in which an external or storage substrate is utilized.

The biomass equation for the chloroplast appears to be missing critical components. Where is the DNA/RNA present in the plastid? Do the authors allow for chloroplast proteins to be important from the cytosol? A major fraction of the biomass of chloroplasts comes from nuclear encoded proteins.

The authors should be more certain about the result presented in line 345. Can't one remove the pigment synthesis unique to the P and C models and see if the growth rates become equal?

What do you mean by line 378 "at best"?

What is incorrect about models regarding proton pumping (line 384)? FBA doesn't need concentrations, only that the stoichiometry is correct.

Figure 2 is too difficult to read.

Figure 3 and 4 could be represented by just 2 numbers representing the slopes. There are no shifts in slope over different photon levels. In fact Figure 4 seems redundant with 3.

How do the results in Fig. 3 and 4 compare to photons/chloroplast? chlorophylls/chloroplast? In other words, how much light is absorbed by each g DW? Is it the same for each organism? Line 408 reads poorly. Photons aren't generated

The line 455 where the model is forced to produce NADPH and ATP in a specific ratio makes the result on line 458-459 meaningless. Of course the ration needed by the CBB is met. Could the authors comment for splitting 1 H20 how many NADPH are made and how many protons are released and what stoichiometry of the ATP synthase of protons per ATP is used.

Figure 5 the y-axis is a growth rate, but the text (line 500) says it should be lipid production.

Most of the conclusions are not conclusions; e.g 518-520, 520-322 are basic statements true without the work in the manuscript. The rest of the conclusions are just a restatement or summary of result.

A minor recommendation (not required) on the tags that would make the model more generalizable is that "C" is not specific enough as many algae start with C, like Chlorella. A three or four letter abbreviation would be more appropriate.

Is the model capable of simulating chromoplasts as well as chloroplasts?

Don't start sentences with #. e.g. line 321

Line 486 please check the wording-"combined pool important chloroplast-produced" the word important doesn't makes sense. Do you just mean total lipids?

6. PLOS authors have the option to publish the peer review history of their article (what does this mean?). If published, this will include your full peer review and any attached files.

Reviewer #1: Yes: John A Morgan

---

## [Author Response · Author response to Decision Letter 0]

8 Jan 2020

We would like to thank Dr. Morgan for his comments and suggestions, and for taking the time and effort to review our manuscript. We have responded to his comments below (responses are marked >>>), and we hope that our revised manuscript will be considered suitable for publication.

Comments from Reviewer:

The manuscript as written provides little new insight into the metabolic fluxes/metabolism taking place within chloroplasts of algae. A lot of statements are obvious or text book level. For example line 359-372, 410-431, 489-493 are all basic knowledge of photosynthesis.

>>> We apologize for leaving the reader with the expectation that novel insights should be gained in this section. Indeed, the textbook level statements are intended to describe the detailed, known processes that are in fact included in our model. A challenge with previous models has been that they have been lacking in this respect. Our primary focus has been to develop a tool that will aid model building of organisms that contain chloroplasts, and to offer a new approach to inclusion of organelles into an existing model. We indeed emphasize in the manuscript that this model is a proof of principle, offering a new approach to model building, and that it is meant to be expanded by the future users of our model module. For this purpose, we included tools for expanding the model with new reactions in the toolbox found in S2 File (addRxFromModelSyntaxGenerator.m, pickKEGGrx.m and rxGenerator.m).

Consequently, including basic knowledge of photosynthesis in the manuscript is deliberate on our part, since we assume that the main interest in our manuscript will come from scientists working with modelling, and not necessarily from plant biologists. We also put a special emphasis on photosynthesis because we see that the photosynthetic processes are often incorrectly modelled in genome-scale metabolic reconstruction of photosynthetic organisms. It thus seems that even though this knowledge is on a textbook level, it has not found its way from the textbooks and into the published genome-scale reconstructions. We have now made our intentions clearer by including a statement about our intent on lines 86 – 93 and 95 – 102.

The authors should add import of substrates from the cytosol to the model. This is how metabolism actually takes place within algae. This also would allow simulation of dark conditions in which an external or storage substrate is utilized.

>>> We very much agree that import from cytosol to the chloroplast module is important. In fact, most of the substrates used by the chloroplast model are imported from cytosol. In order to differentiate between metabolites belonging to the cytosol and those of the chloroplast module, our model thus also contains a cytosol version of the metabolites that are either imported or exported from the chloroplast. When the chloroplast module is plugged into an exo-model, the script being responsible for joining the models (plugAndPlay.m) searches for these metabolites in the cytosol of the exo-model, and translates the metabolites exported from and/or imported to the chloroplast model into the namespace of the exo-model. This is done to ensure that the transport metabolites are indeed able to be transported between the cytosol of the exo-model and the chloroplast model, and to ensure that the exchange between the chloroplast module and the exo-model is transparent. We also consider this approach very helpful for debugging purposes. 

The only 'metabolites' that are not imported from cytosol are the photons used by in the photosynthetic reactions, since light is not a conventional metabolite being transported between organelles. Importing photons directly to the chloroplast also ensures that the combined model consisting of the exo-model and the chloroplast model will run, even if the exo-model does not import photons. We have now added a statement to clarify this issue on lines 162 – 172.

The biomass equation for the chloroplast appears to be missing critical components. Where is the DNA/RNA present in the plastid? Do the authors allow for chloroplast proteins to be important from the cytosol? A major fraction of the biomass of chloroplasts comes from nuclear encoded proteins.

>>> We thank the Reviewer for bringing up this observation. Synthesis of RNA and DNA has been omitted in the current version of the model because these reactions are not essential to chloroplast function. Instead, we have focused on the synthesis of lipids, pigments and proteins, since these are metabolites that are currently attracting a lot of scientific interest. Certain fatty acids that are only synthesized in the chloroplast are transported to other parts of the cell and used further, and the chloroplast also synthesizes several essential amino acids. The generated lipids and proteins / amino acids are therefore metabolite groups that the chloroplast needs to produce in order for the combined model to run, while the same cannot be said about RNA and DNA.

The model in its current form is intended as a proof of principle. Specific reactions related to production of RNA and DNA that would be of interest for e.g. a particular exo-model can of course be added via the tools we provide for adding new reactions to the model. We have now added a statement to explicitly address this point on lines 134 – 140.

The authors should be more certain about the result presented in line 345. Can't one remove the pigment synthesis unique to the P and C models and see if the growth rates become equal?

>>> We thank the Reviewer for bringing up this interesting point. Even if the organism-specific pigment component was removed from the chloroplast biomass reaction, the Phaeodactylum-specific and the Chlamydomonas-specific growth rates would still not be equal. One of the reasons for this is that Chlamydomonas has an active eyespot, and the metabolism of the eyespot is included in the Chlamydomonas mode, but not in the Phaeodactylum mode. In addition, there are differences in the composition of proteins and lipids. Proteins in Phaeodactylum and Chlamydomonas contain the same 20 amino acids, but in different ratios. The ratio between different fatty acids in the Phaeodactylum and Chlamydomonas specific pools of MGDG, DGDG and SQDG is also different. Both Phaeodactylum and Chlamydomonas are able to produce C22:6 (Nannochloropsis is not), but Chlamydomonas does for example produce C18:3 (now stated in the manuscript, at lines 376 – 378), which is not present in Phaeodactylum (if it is present, it is assumed to be so at negligible amounts). 

Thus, even if pigment synthesis would be removed, the growth rates of Phaeodactylum and Chlamydomonas would therefore still not be equal, as the difference in the ratios of amino acids in proteins, and of fatty acids in different lipid pools will create a shift in the fluxes of individual reactions that would be obvious if the flux vectors resulting from optimizations of the two organisms were compared.

We also state in the manuscript that the composition of lipids and proteins are dissimilar in the different organism modes (lines 378 – 383), and we refer to the exact composition in S4 Text (line 383). 

What do you mean by line 378 "at best"?

>>> We thank the Reviewer for bringing up this unfortunate choice of phrase. We mean that in most published models of photosynthetic organisms, photosynthesis is mostly described by less than ten reactions. We have now changed this phrase in the manuscript to hopefully make our intention clearer (line 431).

What is incorrect about models regarding proton pumping (line 384)? FBA doesn't need concentrations, only that the stoichiometry is correct.

>>> We very much agree with the Reviewer's observation of FBA properties. Here, if the number of protons pumped across the thylakoid membrane is not correct, this does affect the stoichiometry, and in particular the ratio between ATP and NADPH produced during photosynthesis. In several published models of photosynthetic organisms, the ratio between ATP and NADPH produced in photosynthesis is incorrect, and the reason for this is that the net number of protons transported across the thylakoid membrane during the production of one molecule of NADPH is not correct.

Figure 2 is too difficult to read.

>>> We thank the Reviewer for bringing up this point. We have now tried to make the figure more readable: The font size has been increased, and the text has been made bold. In addition, nodes representing ‘metabolites’ and ‘reactions’ have gotten separate colours. We have also marked the reactions taking place in the different photosynthetic complexes. The figure text on lines 529 – 533 has been changed to properly explain the new version of the figure.

Figure 3 and 4 could be represented by just 2 numbers representing the slopes. There are no shifts in slope over different photon levels. In fact Figure 4 seems redundant with 3.

>>> Here, we are of the opinion that a figure may be a more comprehensible way of illustrating a phenotype phase plane, instead of just drawing two slopes. Regarding redundancy of figures 3 and 4; we agree with the Reviewer’s point. We have chosen to only keep the figure labelled figure 4 in the first submission of the manuscript, showing growth rate as a function of photon usage by PSII and the rate of carbon fixation by the Calvin-Benson cycle. This figure has now been renamed Fig 3 (lines 458 and 535), and the old figure 5 has been renamed figure 4 (lines 570, 574 and 581).

How do the results in Fig. 3 and 4 compare to photons/chloroplast? chlorophylls/chloroplast? In other words, how much light is absorbed by each g DW? Is it the same for each organism?

>>> In both Nannochloropsis, Phaeodactylum and Chlamydomonas mode, photosynthesis is running at the maximum allowed flux for photon import, meaning that 1000 mmol of photons is imported to the chloroplast per gram dry weight per hour. We have now included a statement to clarify this point on lines 467 – 470.

Line 408 reads poorly. Photons aren't generated

>>> We thank the Reviewer for catching this unfortunate statement. We have now corrected this sentence in the revised version of the manuscript (lines 465 – 466).

The line 455 where the model is forced to produce NADPH and ATP in a specific ratio makes the result on line 458-459 meaningless. Of course the ration needed by the CBB is met. Could the authors comment for splitting 1 H20 how many NADPH are made and how many protons are released and what stoichiometry of the ATP synthase of protons per ATP is used.

>>> We thank the Reviewer for bringing up this point. The reason for restricting the rate of cyclic electron transport was that the photosynthetic processes did not produce NADPH and ATP in the ratio that the Calvin-Benson cycle needed -- photosynthesis did, in fact, not produce NADPH at all. This is why we saw that it might be necessary to restrict the rate of cyclic electron transport, and we built the mini-model consisting only of the photosynthetic electron transport chain and the Calvin-Benson cycle in order to find the most sensible upper bound for the reaction being responsible for cyclic electron transport.

1 molecule of NADPH is created per H2O molecule splitted. Simultaneously, 12 protons are pumped into the thylakoid lumen during one complete S-cycle, meaning that the H+:H2O ratio is 6:1. Regarding ATP synthase, most ATP synthases require 12 protons to be pumped from the lumen and back to the stroma in order for its γ subunit to make a 360 degree movement, which will regenerate 3 molecules of ATP, yielding an ATP:H+ ratio of 1:4. These ratios can also be calculated from table S7. In addition, we state in the manuscript that when cyclic electron transport is eliminated, the photosynthetic electron transport chain produce the exact ratio of ATP and NADPH that the Calvin-Benson cycle needs (lines 521 – 523), and on lines 475 – 477 we state that the ATP:NADPH ratio needed by the Calvin-Benson cycle is 9:6.

Figure 5 the y-axis is a growth rate, but the text (line 500) says it should be lipid production.

>>> We thank the Reviewer for catching this mistake. We have now changed the y-axis label in the figure (now figure 4) to 'rate of lipid production'.

Most of the conclusions are not conclusions; e.g 518-520, 520-322 are basic statements true without the work in the manuscript. The rest of the conclusions are just a restatement or summary of result.

>>> We apologize for this, and we have now expanded the conclusion (lines 587 – 603, 609, 615, 617 – 620 and 784 - 794). We have now tried to emphasize what is new about our model, and what the advantages of organelle modules such as our chloroplast module are. We have also included a few extra references, to add some context, and to give an example of other approaches being used by the modelling community to represent differences in metabolism between cells. 

A minor recommendation (not required) on the tags that would make the model more generalizable is that "C" is not specific enough as many algae start with C, like Chlorella. A three or four letter abbreviation would be more appropriate.

>>> We agree with the Reviewer, especially since several of the tools we have developed are meant to facilitate further expansion of the model by adding new reactions and new organism modes. We have now changed the tags of the model into Nan for Nannochloropsis, Pha for Phaeodactylum, and Chl for Chlamydomonas. The organism tags has been changed both in the reaction vector of the model, in the four scripts using organism tags (addRxFromModelSyntaxGenerator.m, changeOrganismMode.m, pickKEGGrx.m, and rxGenerator.m (All scripts are found in S2_File), and in the explanation of the organism tags in the manuscript (lines 121 – 123).

Is the model capable of simulating chromoplasts as well as chloroplasts?

>>> To our knowledge, chromoplasts are mainly associated with storage of pigments, while photosynthesis does not play a big part. Especially in the chromoplasts found in plant roots, we assume that photosynthesis is absent. We put a lot of emphasis on the photosynthetic electron transfer chain during model development. Yet, if the chloroplast model is plugged into an exo-model that does not require it to produce ATP and NADPH photosynthetically, we see no reason why a chromoplast organism mode could not be introduced to the chloroplast model. Synthesis reactions of several pigments are already present in the chloroplast module, and other synthesis pathways could be introduced in a new organism mode. For getting the chloroplast module to store metabolites, a common solution would be to introduce a direct “exchange” reaction (transport out of the cell) for each of the metabolites to be stored – thus not breaking the steady-state assumption.

Don't start sentences with #. e.g. line 321

>>> We thank the Reviewer for this comment, and have now changed the beginning of these sentences in the revised manuscript (lines 360, 364 – 365, 367, 369 – 370 and 372).

Line 486 please check the wording-"combined pool important chloroplast-produced" the word important doesn't makes sense. Do you just mean total lipids?

>>> We thank the Reviewer for catching this mistake. It seems the sentence was missing a word. This has now been corrected (line 561).

---

## [Editor Report · Decision Letter 1]

6 Feb 2020

An adjustable algal chloroplast plug-and-play model for genome-scale metabolic models

PONE-D-19-26969R1

Dear Dr. Bjerkelund Røkke,

We are pleased to inform you that your manuscript has been judged scientifically suitable for publication and will be formally accepted for publication once it complies with all outstanding technical requirements.

With kind regards,

Andrew Webber

Academic Editor

PLOS ONE

Additional Editor Comments (optional):

Thank you for your careful set of responses.
---

## [Editor Report · Acceptance letter]

10 Feb 2020

PONE-D-19-26969R1 

An adjustable algal chloroplast plug-and-play model for genome-scale metabolic models 

Dear Dr. Bjerkelund Røkke:

I am pleased to inform you that your manuscript has been deemed suitable for publication in PLOS ONE. Congratulations! Your manuscript is now with our production department. 

With kind regards,

on behalf of

Dr. Andrew Webber 

Academic Editor

PLOS ONE